# In Underweight Women, Insufficient Gestational Weight Gain Is Associated with Adverse Obstetric Outcomes

**DOI:** 10.3390/nu15010057

**Published:** 2022-12-23

**Authors:** Alizée Montvignier Monnet, Delphine Savoy, Lise Préaubert, Pascale Hoffmann, Cécile Bétry

**Affiliations:** 1Midwifery Department, Faculty of Medicine, University Grenoble Alpes, CHU Grenoble Alpes, 38000 Grenoble, France; 2Département de Gynécologie-Obstétrique & Médecine de la Reproduction, Centre Hospitalier Universitaire Grenoble Alpes (CHUGA), La Tronche, France, Institut National de la Santé et de la Recherche Médicale U1292, Biologie et Biotechnologie pour la Santé, 38000 Grenoble, France; 3Univ. Grenoble Alpes, CNRS, UMR 5525, VetAgro Sup, Grenoble INP, DepartCHU Grenoble Alpes, TIMC, 38000 Grenoble, France

**Keywords:** pregnancy, newborn, obstetric outcome, birth weight, foetal growth restriction, thinness

## Abstract

The pre-pregnancy BMI and the gestational weight gain are two important determinants of pregnancy outcomes. The aim of this study was to determine obstetric outcomes associated with insufficient gestational weight gain in women with a pre-pregnancy BMI < 18.5 kg/m^2^. This study was based on observational routinely collected data from University Hospital Maternity. The participants were allocated to the group sufficient or insufficient gestational weight gain: ≥12.5 kg and <12.5 kg respectively. Primary outcomes were the adjusted birth weight in percentiles (%) and the proportion of SGA newborns. Secondary outcomes were obstetric and perinatal outcomes. A total of 132 participants with a median age of 28 ± 8 years were included. The adjusted birth weight in percentiles was significantly lower in the insufficient gestational weight gain group (27.3 ± 45.0 vs. 46.3 ± 46.2%; *p* < 0.001). Moreover, the insufficient gestational weight gain is associated with a higher risk of SGA (27.0% vs. 11.6%; *p* = 0.03). Our study also showed increased risks of premature rupture of membranes, anaemia, and intrauterine growth restriction in women with an insufficient weight gain. Future studies should explore the risk factors associated with insufficient weight gain, in order to develop specific care for underweight pregnant women.

## 1. Introduction

The pre-pregnancy BMI and the gestational weight gain are two important determinants of pregnancy outcomes. In particular, there is a strong link between gestational weight gain and newborn birth weight [1]. According to the US Institute of Medicine (US IOM) guidelines, and National Research Council 2009, the optimal gestational weight gain depends on the pre-pregnancy BMI [2]. The gestational weight gain is out of the recommendations in the majority of women, with 23% presenting with an insufficient gestational weight gain and 47% with excessive weight gain. Insufficient gestational weight gain is associated with higher risks of prematurity and small for gestational age [3].

To date, the majority of studies about pre-pregnancy BMI have focused on overweight and obese women. Little research has been done on the impact of being underweight before pregnancy on obstetric outcomes. It has been shown that a pre-pregnancy BMI below 18.5 kg/m^2^ is associated with a significant increase in the risk of prematurity and small for gestational age [4,5,6].

There is still uncertainty about the impact of sufficient or insufficient gestational weight gain in underweight women. We hypothesized that an insufficient gestational weight gain (below 12.5 kg according to the IOM recommendations) is associated with adverse obstetric and perinatal outcomes in this population. The main objective of this study was to compare the adjusted birth weight in percentiles (%) and the proportion of small for gestational age newborns in women with pre-pregnancy BMI < 18.5 kg/m^2^ and sufficient or insufficient gestational weight gain. We also sought to study the impact of insufficient gestational weight gain on obstetric and perinatal outcomes.

## 2. Materials and Methods

### 2.1. Study Population

This was a retrospective cohort study at the University Hospital Maternity of Grenoble, France. All women who delivered between April 2018 and May 2019 were screened. It was based on observational routinely collected data.

#### 2.1.1. Inclusion Criteria

All women aged 18 and older, with a pre-pregnancy BMI < 18.5 kg/m^2^ and who delivered a single baby after 24 weeks of amenorrhea during the study period were included.

#### 2.1.2. Exclusion Criteria

Cases of medical termination, participants with significant active comorbidities (such as neoplasia, autoimmune diseases including pre-existing diabetes, uterine malformations) or missing data (birth weight or gestational weight gain) were excluded.

#### 2.1.3. Group Allocation

IOM guidelines recommend a weight gain between 12.5 and 18 kg in pregnant women with a pre-pregnancy BMI < 18.5 kg/m^2^ [2]. Accordingly, the participants were allocated to the group “sufficient gestational weight gain; ≥12.5 kg” or “insufficient gestational weight gain; <12.5 kg” based on their gestational weight gain.

### 2.2. Collection of Data

Data were retrospectively collected from electronic medical records.

#### 2.2.1. Characteristics of the Participants

The maternal characteristics included pre-pregnancy weight (kg), height (cm), maternal age at delivery (years), weight on the day of delivery (kg), parity, occurrence of hyperemesis gravidarium, eating disorders according to the Diagnostic and Statistical Manual of Mental Disorders Fifth Edition (DSM-V), and any toxic exposure including tobacco during pregnancy (smoking, alcohol, or illegal drugs use). 

In our maternity department, the weight before pregnancy is collected declaratively at the first antenatal visit. The weight on the day of delivery was sporadically available. Thus, for all participants, we estimated the total weight gain in kg with the following formula:(1)Weight at the last antenatal visit+0.5×(number of weeks between the delivery and the last antenal visit)

Indeed, the IOM guidelines estimate the rate of weight gain at 0.5 kg per week in underweight women in the third trimester [2].

#### 2.2.2. Main Outcomes

Primary outcomes were the adjusted birth weight in percentiles (%) and the proportion of small for gestational age (SGA) newborns. The birth weight (kg) was measured in the delivery room. The adjusted birth weight in percentiles was determined using the Audipog application. This score-free application is based on the gestational age at birth and sex of the child [7]. It allows adjusting data on the gestational length and is based on French data. The threshold used for defining a newborn SGA was <10%.

Secondary outcomes were based on obstetric complications: hypertensive disorder of pregnancy (systolic blood pressure higher than 140 mmHg and/or a diastolic blood pressure higher than 90 mmHg on at least two occasions), gestational diabetes (hyperglycaemia first diagnosed during pregnancy identified according to the International association of diabetes and pregnancy study group guidelines [8]), anaemia (haemoglobin level at the 6th month < 10.5 g/L), premature rupture of membranes, threatened preterm labour (uterine contractions and ultrasound cervical length < 25 mm before 37 weeks of gestation), intrauterine growth restriction (foetal weight estimated gestational age below the 3rd percentile or below the 10th percentile with oligohydramnios or doppler abnormalities or decreased foetal movement), gestational age at delivery (weeks), preterm delivery (<37 weeks), caesarean delivery, and newborn characteristics (umbilical artery pH, Apgar Score < 7 at 5 min, sex).

### 2.3. Ethics Statement

Under French law, this study is exempt from institutional review board review because it is an observational study using anonymized data from medical records. Women were informed in the institutional leaflet that their records can be used for the evaluation of medical practices and are informed that they can opt out of these studies (MR-004 reference methodology) [9]. 

### 2.4. Statistical Analysis

Sample size calculation was performed using BiostaTGV (https://biostatgv.sentiweb.fr/?module=etudes/sujets# accessed on 18 September 2019). It was based on a frequency of SGA of 14.2% in women with sufficient gestational weight gain and 31.3% in women with insufficient gestational weight gain and pre-pregnancy BMI < 18.5 kg/m^2^ [3]. Setting the power at 80% and the one-sided significance level at 0.05, 72 individuals were required in each group. Descriptive data are presented as median ± interquartile (IQR) or frequencies (%). Non-parametric statistics were used. The Mann-Whitney test was used for comparisons of continuous variables. The Fischer exact test was used for comparisons of proportions of categorical variables. Given an age difference between both groups, subsequent comparisons were assessed by logistic or multiple linear regression adjusted for age. Statistical analysis was performed using Jamovi software (Version 1.16.23.0). The *p*-values <0.05 were considered significant.

## 3. Results

### 3.1. Characteristics of the Included Participants

Participants meeting inclusion criteria (*n* = 194) were screened. A total of 132 participants with a median age of 28 ± 8 years were included in the insufficient (*n* = 63) and sufficient (*n* = 69) gestational weight gain groups (Figure 1). 

As expected, the maternal weight at delivery and the gestational weight gain were lower in the insufficient gestational weight gain group vs. sufficient (59 ± 6 vs. 65 ± 9 kg and 10.5 ± 2.0 vs. 16.0 ± 5.0 kg, respectively; *p* < 0.001 for both comparisons). The maternal age was higher in the insufficient gestational weight gain group vs. sufficient (29 ± 10 vs. 27 ± 8 years; *p* = 0.04). There were no significant differences in pre-pregnancy BMI between both groups. There were no significant differences in potential pregnancy risk factors for insufficient gestational weight gain between both groups (Table 1).

### 3.2. Primary Outcome: Adjusted Birth Weight and Percentages of SGA Newborns

Median birth weight was significantly lower in the insufficient gestational weight gain group vs. sufficient (2980 ± 615 g vs. 3300 ± 620 g, respectively; *p* < 0.001) as well as the birth weight in percentiles adjusted on sex and the gestational age at birth (Figure 2). Insufficient gestational weight gain was significantly associated with a higher risk of SGA: 17 (27.0%) vs. 8 (11.6%) in women with a sufficient gestational weight gain; *p* = 0.03).

### 3.3. Secondary Outcomes

Table 2 summarizes the pregnancy outcomes and the characteristics of the newborns. Insufficient gestational weight gain was significantly associated with an increased risk of anaemia, premature rupture of membranes, intrauterine growth restriction and lower gestational age at delivery.

## 4. Discussion

The impact of pre-pregnancy obesity and excessive weight gain during pregnancy are well known. In contrast, few studies have addressed the questions of consequences of both a pre-pregnancy BMI <18.5 kg/m^2^ and an insufficient gestational weight gain. In this study, we aimed to determine the impact of an insufficient gestational weight gain on obstetric and neonatal outcomes in underweight women.

As far as we know, this is the first study in France on this topic, while majority of the studies were done in the Unites States or in Asia [3]. We demonstrated that in underweight women, an insufficient gestational weight gain was associated with a higher risk of SGA after adjustment on gestational age and sex. This is in agreement with previous studies including a meta-analysis [3,10,11,12,13,14]. We also demonstrated that an insufficient weight gain was associated with an increased risk of intrauterine growth restriction. It has been previously shown that, on the one hand, women with a BMI < 18.5 kg/m^2^ have a higher risk of intrauterine growth restriction [15] and, on the other hand, women with a low gestational weight gain are also known to be at risk for intrauterine growth restriction [16]. However, to the best of our knowledge, this is the first study focusing on the clinical impact of the gestational weigh gain in underweight women. The fact that an insufficient weight gain is associated with an increased risk of intrauterine growth restriction is clinically important. Underweight women and healthcare professionals ought to be informed of that risk. As this result is based on a secondary outcome analysis, it should be confirmed by further studies.

We also showed that insufficient gestational weight gain was associated with obstetrical complications in underweight women. Indeed, we observed an increased risk of premature rupture of membranes in our insufficient gestational weight gain group. This finding contrasts with a previous study of Taiwanese women [14]. Apart from this discordance, our results are in agreement with published studies showing an increased risk of preterm birth related to premature rupture of membranes in case of insufficient weight gain, independent of the BMI [17], and in Japanese underweight women [11]. In our study, we find an increased risk of prematurity (but not significative), and a lower gestational age at delivery. Lastly, the risk of anaemia was shown to be higher in case of insufficient weight gain. This result corroborates previous studies showing that anaemia is more frequent in case of insufficient gestational weight gain regardless of BMI [18,19,20]. As far as we know, no other authors have studied the occurrence of anaemia in underweight women depending on their gestational weight gain.

A number of potential limitations related to the retrospective nature of our study need to be considered. First, participants were usually not weighed on the day of delivery. We decided to add 0.5 kg for each week between the last antenatal visit and the delivery according to the IOM recommendations. Nevertheless, some women could have gained more or less weight, especially those with insufficient weight gain. Weight at the delivery is not always reported in previous studies, and in some cases the weight at the last antenatal visit is considered as the final weight [3]. We believe that our method is more appropriate to ensure homogenate collection of data. Second, given the retrospective design, some data were missing, and we cannot be sure that some risks factors associated with insufficient weight gain were correctly mentioned in the electronic medical record. Thus, we cannot conclude that eating disorders, or toxic exposure during pregnancy are not risk factors for insufficient weight gain. Finally, the size of the study population is relatively small. We calculated the sample size according to the main objective of the study, but we are aware that the lack of significance for secondary objectives could be related to the insufficient power of our study.

Our findings might be useful for clinicians following underweight pregnant women. Following the IOM recommendations in French underweight women may help to prevent the risk of SGA and increase the birth weight. Further work should focus on the factors associated with an insufficient weight gain in underweight women. Especially, it would be of interest to determine the trajectory of weight gain depending on the aetiology of underweight, e.g., constitutive thinness, anorexia nervosa, or orthorexia. A recent systemic review suggested that active anorexia nervosa could be associated with adverse pregnancy outcomes including prematurity, intrauterine growth restriction, or low birth weight [21]. In a case series, it was shown that women with active anorexia nervosa had a lower weight gain and newborns with lower birth weight than women in remission from eating disorders [22]. Identifying women with eating disorders during the prenatal and antenatal visits may help to identify women at risk for insufficient weight gain. Another important factor could be the presence of gestational diabetes. In France, the prevalence of gestational diabetes is estimated at 8%. And, it is well known, that a higher BMI is associated with a higher risk of gestational diabetes [23]. Thus, in our study, the prevalence of gestational diabetes is greater than expected in the group of women with insufficient gestational weight gain (12%). It has been shown that gestational diabetes is associated with insufficient gestational weight gain in 50,3% of cases [24]. Even if future studies on this topic are required to validate the links between insufficient gestational weight gain, gestational diabetes, and poor outcomes in women with pre-pregnancy < 18.5 kg/m^2^, practitioners should be mindful of these potential risks, when they advise the patient.

In conclusion, in underweight women, insufficient gestational weight gain is associated with adverse obstetric outcomes including a lower adjusted weight birth as well as an increased risk of SGA. Our study also shows an increased risk of premature rupture of membranes, anaemia, and intrauterine growth restriction. In order to develop specific care for underweight pregnant women, future studies should explore the risk factors associated with insufficient weight gain.

## Figures and Tables

**Figure 1 nutrients-15-00057-f001:**
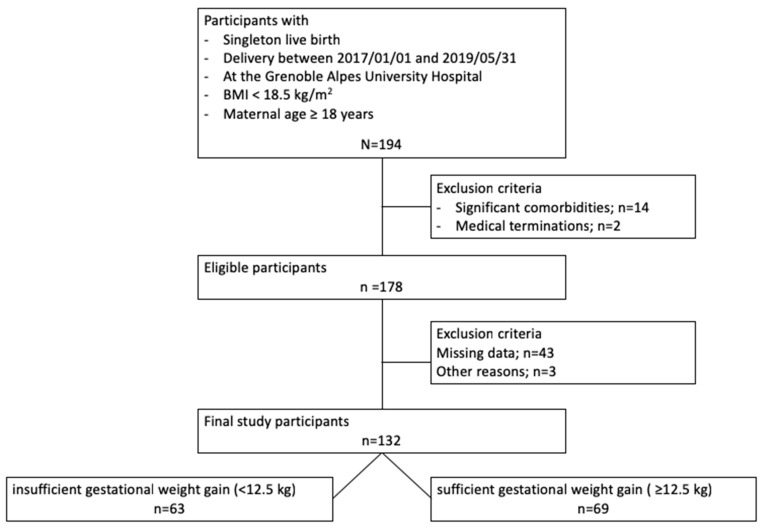
Flowchart of the study; BMI: body mass index.

**Figure 2 nutrients-15-00057-f002:**
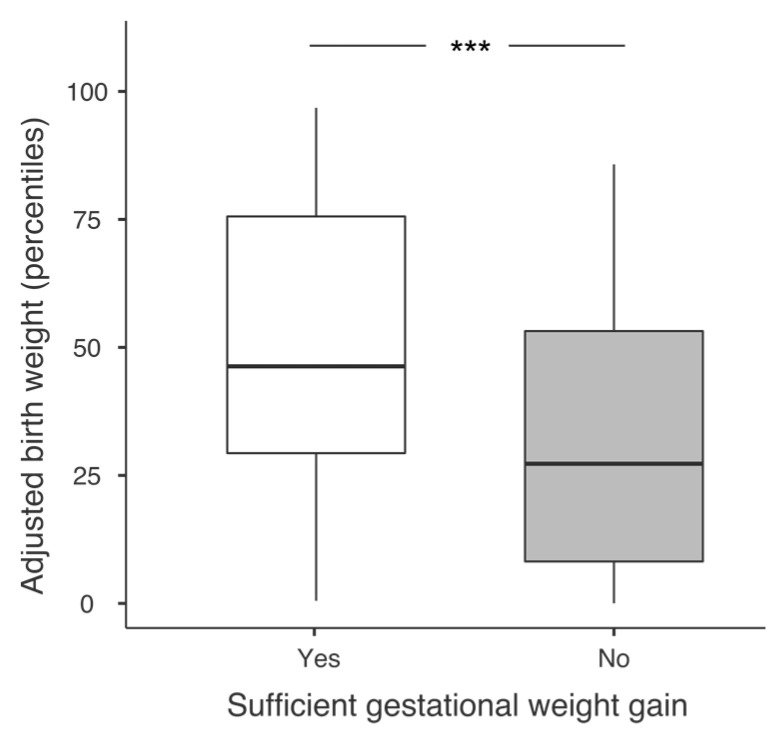
Boxplots for median adjusted birth weight in percentiles according to the gestational weight gain group (sufficient (≥12.5 kg) or insufficient (<12.5 kg) gestational weight gain). *** *p* < 0.001 using multiple linear regression adjusted for maternal age.

**Table 1 nutrients-15-00057-t001:** Potential pregnancy risk factors of insufficient weight gain in median (IQR) or in number (frequency) according to the status of sufficient (≥12.5 kg) or insufficient (<12.5 kg·m^−2^) gestational weight gain.

	Sufficient Gestational Weight Gain (*n* = 69)	Insufficient Gestational Weight Gain (*n* = 63)	*p* Value
Primipara	31 (45%)	26 (41%)	*p* = 0.7
Hyperemesis gravidarium	3 (4%)	1 (2%)	*p* = 0.6
Eating disorders	2 (3%)	2 (3%)	*p* = 1
Toxic exposure during pregnancy			
Smoking	18 (27%)	17 (28%)	*p* = 1
Alcohol use	0 (0%)	1 (2%)	*p* = 0.5
Illegal drugs use	2 (3%)	3 (5%)	*p* = 0.7

**Table 2 nutrients-15-00057-t002:** Obstetric and neonatal outcomes in median (IQR) or in number (frequency) according to the status of sufficient (≥12.5 kg) or insufficient (<12.5 kg·m^−2^) gestational weight gain.

	Sufficient Gestational Weight Gain (*n* = 69)	Insufficient Gestational Weight Gain (*n* = 63)	*p* Value
Pregnancy outcomes			
Hypertensive disorder of pregnancy	3/68 (4%)	1/63 (2%)	*p* = 0.5
Gestational diabetes	3/68 (4%)	7/61 (12%)	*p* = 0.3
Anaemia	18/58 (31%)	23/46 (50%)	*p* = 0.026
Premature rupture of membranes	13/68 (19%)	25/60 (42%)	*p* = 0.008
Threatened preterm labor	9/69 (13%)	11/63 (18%)	*p* = 0.4
Intrauterine growth restriction	3/69 (4%)	10/59 (17%)	*p* = 0.036
Gestational age delivery (weeks)	40.1 (1.5)	39.3 (2.2)	*p* = 0.017
Preterm delivery (<37 weeks)	4/69 (6%)	8/63 (13%)	*p* = 0.2
Caesarean delivery	7/69 (10%)	7/63 (11%)	*p* = 0.8
Newborns characteristics			
Umbilical artery pH	7.26 (0.09)	7.29 (0.10)	*p* = 0.08
Apgar Score < 7 (at 5 min)	0/69 (0%)	3/63 (5%)	*p* = 1
Male infants (%)	29/69 (42%)	31/63 (49%)	*p* = 0.3

For some outcomes, there were missing data.

## Data Availability

Data will be available upon request.

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
