# Peer review of "In Underweight Women, Insufficient Gestational Weight Gain Is Associated with Adverse Obstetric Outcomes"

_nutrients, 2022, doi:10.3390/nu15010057_

Round 1
Reviewer 1 Report
1. Did the collected data (patient demographics) include race?
2. Participants with significant ‘’active” comorbidities were excluded. What about those with chronic –but not active –comorbidities (peripheral vascular disease, chronic renal disease...etc) ?
3. Data about the neonatal outcomes highlighted in relation to the mode of birth in the study groups?
4. Was there any correlation between different maternal age groups and SGA in the studied population?
5. Socioeconomic status of population studied-any information ?
6. Delivery outcomes between primipara and multipara in study group –is not so clear
Author Response
Response to Reviewer 1 comments :
1. Did the collected data (patient demographics) include race?
Response : No, French law does not authorize the collection of information about race.
2. Participants with significant ‘’active” comorbidities were excluded. What about those with chronic –but not active –comorbidities (peripheral vascular disease, chronic renal disease...etc) ?
Response : We excluded patients if we considered that the comorbidity could impact the weight or the nutritional status of the participant
3. Data about the neonatal outcomes highlighted in relation to the mode of birth in the study groups?
Response: Thank you for pointing this out. Although we agree that this is an important consideration, it is beyond the scope in this manuscript. The topic of our study was to assess the impact of gestational weight gain on obstetric outcomes. Since, there was no significant differences of caesarian delivery between the two groups (sufficient and insufficient gestational weight gain), caesarian delivery was not considered as a confusion factor for neonatal outcomes.
4. Was there any correlation between different maternal age groups and SGA in the studied population?
Response: Thank you for your question. As mentioned in the methods, "Given an age difference between both groups, subsequent comparisons were assessed by logistic or multiple linear regression adjusted for age". Thus, the risk of SGA mentioned in the results is adjusted on maternal age.
5. Socioeconomic status of population studied-any information ?
Response: Although we agree that this is an important consideration, it cannot be analyzed in this manuscript because the information was not available in electronical medical records
6. Delivery outcomes between primipara and multipara in study group –is not so clear
Response: Thank you for pointing this out. As mentioned above, we think it is also beyond the scope in this manuscript. The topic of our study was to assess the impact of gestational weight gain on obstetric outcomes. Since, there was no significant differences of parity between the two groups (sufficient and insufficient gestational weight gain), parity was not considered as a confusion factor for delivery outcomes.
Reviewer 2 Report
Dear author's
I was pleased to review your article and I have the following comments:
1. The study is retrospective and for this reason there is a risk of bias.
2. The sample is relatively small.
3. What do you think about a control group?
4. Please explain the limitation of your study.
5. In the section discussion it is mandatory to compare your results with the existing literature.
Author Response
Response to Reviewer 2.
1. The study is retrospective and for this reason there is a risk of bias.
Response: We agree that this is a potential limitation of the study. We mentioned this as a limitation on the discussion section (lines 416-420)
2. The sample is relatively small.
Response: We agree that this is a potential limitation of the study. We have added this as a limitation on the discussion section (lines 420-423)
3. What do you think about a control group?
Response: The main objective of our study was to compare sufficient and insufficient gestational weight gain in underweight women. For this reason, the control group was the group with sufficient gestational weight gain in our study. We agree that comparing insufficient weight gain in patients with normal BMI can give other important information, but it was not in the scope of our study.
4. Please explain the limitation of your study.
Response: As mentioned above, we have expanded the paragraph on the limitations of the study in the discussion section
5. In the section discussion it is mandatory to compare your results with the existing literature.
Response: While we appreciate the reviewer’s feedback, we consider that we have already compared our results with the literature in the second and third paragraphs of the discussion, as expected.
Reviewer 3 Report
This retrospective French study analyses the role of weight gain during pregnancy in women with pre pregnancy BMI less than 18,5. I have the following comments:
1. What is the rate of overweight, obese and underweight pregnant women in France?
2. What is the number of births in the authors’ institution each year?
3. Why did the authors select the period between January 2017 and May 2019? Was this selected according to sample size calculation? Could some additional information be derived from a bigger sample?
4. Was only one measurement of blood pressure above 140/90 defined as gestational hypertension?
5. What is the definition of established and threatened preterm labour? Is cervical length measured in all women with uterine contractions before the 37th week in the authors’ institution?
Author Response
Response to reviewer 3
What is the rate of overweight, obese and underweight pregnant women in France?
Response: According to the last national perinatal survey (2021), the rate of overweight, obese and underweight pregnant were 23%, 13.4% and 5.8% respectively.
What is the number of births in the authors’ institution each year?
Response: About 2800 births each year
Why did the authors select the period between January 2017 and May 2019? Was this selected according to sample size calculation? Could some additional information be derived from a bigger sample?
Response: Thank you for pointing this out, it helps us to identify a mistake in the database. According to the number of births in our institution and the prevalence of underweight women, we defined a period of inclusion of one year, which was secondary extended to 13 months given the sample size calculation. Three women give birth during the included period but also in 2017. For these 3 women, the data were collected by mistake for the birth in 2017 and not in the inclusion period. We decided to remove these three women in the database and to modify the manuscript accordingly. It does not change neither the results nor the conclusion of the study. Furthermore, we agree that the small size of the study population is a potential limitation of the study. We have added this as a limitation on the discussion (lines 420-423)
Was only one measurement of blood pressure above 140/90 defined as gestational hypertension?
Response: Thank you for pointing this out. Hypertensive disorder of pregnancy is defined by an elevated blood pressure on at least two occasions. It has been corrected (line 106)
What is the definition of established and threatened preterm labour? Is cervical length measured in all women with uterine contractions before the 37th week in the authors’ institution?
Response: Threatened preterm labor is defined by contractions without cervical modifications, while established preterm labour drives to birth within few hours.
We measure cervical length for every woman consulting for contractions before the 37th week of gestation.